

# Clinicopathologic features and prognosis of 71 patients with gastric cancer and disseminated intravascular coagulation

Ling Chen, Jing Lin, Yu Chen, Jiami Yu and Xiaojie Wang

Department of Medical Oncology, Clinical Oncology School of Fujian Medical University,
Fujian Cancer Hospital, Fuzhou, Fujian, People's Republic of China

## ABSTRACT

**Background:** Gastric cancer consists of solid tumors with a tendency for disseminated intravascular coagulation (DIC). DIC is rare in patients with stomach cancer, and there have been few studies on this condition. We aimed to perform comprehensive analyses of the prognosis and clinicopathologic characteristics of stomach cancer patients with DIC.

**Methods:** Between June 2006 and March 2020, 14,016 patients at Fujian Cancer Hospital were diagnosed with stomach cancer. We reviewed their medical records and found that 105 of these patients were diagnosed with DIC. After excluding patients who were lost to follow-up, 71 patients with DIC remained. The clinical data were retrospectively analyzed to observe clinical characteristics and prognostic factors, and the Kaplan–Meier survival analysis was performed. Prognostic variables were investigated by the Cox proportional hazards method.

**Results:** The median age was 54 (range, 21–83) years, and 38 patients (53.5%) were male. The histological category was poorly differentiated gastric cancer in 58 patients (81.7%). Eleven patients (15.5%) developed DIC after curative gastric resection. Sixty patients (84.5%) had DIC at the initial presentation of gastric cancer or developed DIC when the tumor progressed during treatment. Fifty-one patients (71.8%) had bleeding symptoms, and 43 (60.6%) patients had comorbidities at the time of DIC diagnosis. Among the 71 patients, 42 (59.2%) had multiple metastatic patterns. Twenty-one (29.6%) patients received chemotherapy. The median overall survival (OS) was 57.0 days (95% confidence interval [CI] [33.1–80.9] days). Tumor status ($P = 0.000$) and treatment ($P = 0.003$) were found to be significant variables associated with OS by univariate analysis. Multivariate analysis showed that tumor status ($P = 0.000$) and treatment ($P = 0.000$) had independent effects on OS.

**Conclusions:** Gastrointestinal bleeding, multiple metastatic patterns and comorbidities at diagnosis with DIC are common in patients with gastric cancer complicated with DIC. Patients with poorly differentiated gastric cancer are more likely to develop DIC. Treatment and tumor status are separate risk variables for the survival of gastric cancer patients with DIC.DIC patients without tumors have a good prognosis and can be cured by appropriate etiological correction and symptomatic treatment. Chemotherapy can improve the prognosis of DIC patients with tumors.

Corresponding author
Ling Chen, chenl05@fjzlhospital.com

## INTRODUCTION

Diffused intravascular coagulation (DIC) is a clinical illness characterized by microcirculatory failure and bleeding. DIC is a pathological process in which pathogenic factors harm the vascular system and trigger coagulation activation, systemic microvascular thrombosis, large consumption of coagulation factors, and secondary hyperfibrinolysis. Severe infection, malignant tumors, pathological obstetric diseases, surgery, trauma, severe poisoning and immune responses are the most common underlying diseases that contribute to DIC (*Levi & ten Cate, 1999*). DIC is also a common hemostatic complication of malignant tumors. According to *Sallah et al. (2001)*, the incidence of DIC was 6.8% among 1117 individuals with solid tumors. *Yamashita et al. (2014)* reported that the incidence of DIC in 478 patients with advanced malignant diseases was 13.2%. Coagulation and fibrinolytic activity are biological characteristics of tumor cells. In patients with malignant tumors, various coagulation factors such as tissue factor and fibrinogen are increased, the physiological anticoagulation pathway is weakened, and fibrinolytic ability is decreased. DIC occurs when the dynamic balance in the hypercoagulable state is disrupted by the inducing factors of DIC, such as radiotherapy, chemotherapy, surgery, and infection (*Donati, 1995*; *Nadir, Vlodavsky & Brenner, 2008*; *Milsom et al., 2008*; *Kwaan & McMahon, 2009*; *Rodeghiero & Castaman, 1994*). DIC has been frequently found in patients with gastric cancer, breast cancer, hepatic cell carcinoma and hematopoietic malignancies. Gastric cancer consists of solid tumors with a tendency for DIC, but there have been few studies on this condition. The majority of these studies (*Chao et al., 2000*; *Hironaka et al., 2000*; *Therasse et al., 2000*; *Tokar et al., 2006*; *Huang et al., 2008*; *Hwang et al., 2014*) are case reports and small-sample studies of 6–68 individuals. DIC can occur in both advanced and early gastric cancer; however, all of the above studies evaluated DIC in advanced gastric cancer (AGC) patients. Seventy-one patients with stomach cancer and DIC were included (60 AGC patients and 11 early gastric cancer (EGC) patients) in this study. The clinical data of these patients were retrospectively analyzed to observe clinical characteristics and prognostic factors.

## MATERIALS AND METHODS

### Medical ethics

This study was approved by the Ethical Committee of the Clinical Oncology School of Fujian Medical University, Fujian Cancer Hospital (permit number K2022-133-01). All individuals taking part in the study gave their written informed consent.

### Patients

We reviewed the medical files of 14,016 patients who received a gastric cancer diagnosis at Fujian Cancer Hospital between June 2006 and March 2020. One hundred and five (0.75%) of these patients were diagnosed with DIC. After excluding patients who were lost to

follow-up, 71 patients with DIC remained. All patients had histologically confirmed gastric cancer, and DIC was diagnosed using the criteria of the International Society on Thrombosis and Hemostasis (ISTH) (*Taylor et al., 2001*). Retrospective assessments of the patients' clinicopathologic characteristics, including sex, age, histological category, tumor status, bleeding symptoms, comorbidities, treatment, and metastatic patterns, were carried out through telephone follow-ups and medical file assessment.

## Clinicopathological factor identification

Patients were divided into two groups by age at the time of DIC diagnosis: <65 years and ≥65 years. Three histological categories were applied: poorly differentiated, highly differentiated and uncertain. Papillary carcinoma and highly and moderately differentiated tubular adenocarcinoma were considered highly differentiated malignancies. Signet ring cell carcinoma, mucinous adenocarcinoma and poorly differentiated tubular adenocarcinoma were considered poorly differentiated malignancies. The uncertain histological category referred to gastric cancer for which no specific pathological classification was given in the pathological report. The tumor status was divided into two categories: tumor free, meaning that DIC occurred after curative gastric resection; and with tumor, meaning that DIC occurred at the initial presentation of gastric cancer or developed as the tumor progressed during treatment. Bleeding symptoms included gastrointestinal bleeding, intracerebral hemorrhage, hemoptysis, petechiae, bloody pleural effusion and ascites, hematuria and gingival bleeding. The comorbidities included simultaneous infection, cerebral infarction and intestinal obstruction at the time of DIC diagnosis. The metastatic patterns included peritoneal implantation (Krukenberg tumors, peritoneal nodules and ascites), hematogenous (bone, lung and liver), locoregional (gastric or nodal), uncertain (unable to confirm the metastatic patterns), and multiple recurrence patterns. Based on the type of treatment, the patients were split into two groups: chemotherapy and best supportive care (BSC). BSC was administered to patients not receiving chemotherapy.

## Postoperative follow-up

Physical examinations, blood tests and routine imaging (chest/abdominopelvic cavity) were all part of the standard follow-up. Patients who exhibited signs of bone metastasis underwent a bone scan. The period from DIC diagnosis to death or the last follow-up was defined as overall survival (OS). The patients were closely monitored until May 2022; the median follow-up time was 138.0 (range, 0–138.0) months.

## Evaluation of tumor response

Based on changes in target and nontarget lesions, objective chemotherapeutic responses were assessed using the Response Evaluation Criteria in Solid Tumors (RECIST) (*Therasse et al., 2000*).

## Statistical analysis

SPSS 21.0 software was used to analyze the data. Comparisons of clinicopathological characteristics between the chemotherapy group and BSC group were performed with the

**Table 1 Clinicopathological characteristics of the 71 patients.**

| Clinicopathological characteristics | $n$ (%) |
|---|---|
| Sex | |
| Male | 38 (53.5) |
| Female | 33 (46.5) |
| Age (years), median (range) | 54 (21-83) |
| <65 | 52 (73.2) |
| ≥65 | 19 (26.8) |
| Histological category | |
| Highly differentiated | 7 (9.9) |
| Poorly differentiated | 58 (81.7) |
| Uncertain | 6 (8.4) |
| Tumor status | |
| With tumor | 60 (84.5) |
| Tumor free | 11 (15.5) |
| Bleeding symptoms | |
| Positive | 51 (71.8) |
| Negative | 20 (28.2) |
| Comorbidities | |
| Negative | 28 (39.4) |
| Positive | 43 (60.6) |
| Metastatic patterns | |
| Uncertain | 1 (1.4) |
| Locoregional metastasis | 12 (16.8) |
| Hematogenous metastasis | 7 (9.9) |
| Peritoneal implantation | 7 (9.9) |
| Multiple metastatic patterns | 42 (59.2) |
| Negative | 2 (2.8) |
| Treatment | |
| BSC | 50 (70.4) |
| Chemotherapy | 21 (29.6) |

chi-squared test or Fisher's exact test. The log-rank test was used to compare survival curves, which were created using the Kaplan–Meier survival analysis method for univariate analysis. Cox regression analysis was used in the multivariate analysis to identify independent clinicopathological characteristics that were related to OS. Statistics were deemed significant at $P < 0.05$. This study used the same statistical methods as other similar studies (*Rhee et al., 2010*; *Hwang et al., 2014*; *Sun et al., 2015*).

# RESULTS

## Patient characteristics

Seventy-one patients with a median age of 54 (range, 21–83) years were enrolled in this study. Table 1 provides an overview of the clinicopathological characteristics of the

patients. Thirty-eight patients (53.5%) were male. The histological category was poorly differentiated gastric cancer in 58 patients (81.7%), highly differentiated in seven patients (9.9%), and uncertain in six patients (8.4%). Eleven patients (15.5%) developed DIC after curative gastric resection. Sixty patients (84.5%) had DIC at the initial presentation of gastric cancer or developed DIC when the tumor progressed during treatment. Fifty-one patients (71.8%) had bleeding symptoms, including gastrointestinal bleeding in 32 patients, intracerebral hemorrhage in four patients, hemoptysis in three patients, petechiae in 16 patients, bloody pleural effusion and ascites in six patients, hematuria in five patients, and gingival bleeding in three patients. Forty-three (60.6%) patients had comorbidities at the time of DIC diagnosis. The comorbidities included simultaneous infection in 42 patients, cerebral infarction in one patient and intestinal obstruction in four patients. Among the 71 patients, 42 (59.2%) had multiple metastatic patterns, 12 (16.8%) had locoregional metastasis, seven (9.9%) had hematogenous metastasis, seven (9.9%) had peritoneal implantation, two (2.8%) had no metastasis, and one (1.4%) had an uncertain metastatic status.

## Treatment

Twenty-one (29.6%) of the 71 patients received chemotherapy. Fourteen of these patients (66.7%) underwent one to three cycles of chemotherapy, while seven (33.3%) underwent four or more cycles. The chemotherapy regimens included paclitaxel every week (47.6%), POF (paclitaxel, 5-fluorouracil (5-FU), oxaliplatin) every 2 weeks (23.9%), TF (paclitaxel, 5-FU) every 2 weeks (9.5%), FOLFIRI (irinotecan, leucovorin, 5-FU) every 2 weeks (9.5%), and paclitaxel plus oxaliplatin every two weeks (9.5%). Five patients in the chemotherapy group had their treatment responses assessed (one patient had a partial response, and four patients had stable disease). Ten patients who developed DIC after curative gastric resection and 40 patients who had DIC at the time of diagnosis or who developed DIC as the tumor progressed during treatment received BSC, including symptomatic treatment such as antibiotic therapy, nutritional support therapy and treatment based on different stages of DIC such as anticoagulation therapy with heparin, antifibrinolytic treatment with tranexamic acid, coagulation factor and platelet transfusions. No significant difference was found in sex, age, histological category, bleeding symptoms, comorbidities, metastatic patterns or tumor status between the chemotherapy group and BSC group ($P > 0.05$) (Table 2).

## Survival analysis

The five-year survival of 71 patients was 2.8%. The median OS was 57.0 days (95% confidence interval [CI] [33.1– 80.9]) (Fig. 1). Tumor status ($P = 0.000$) and treatment ($P = 0.003$) were found to be significant variables linked with OS by univariate analysis (Table 3). The median OS in the with-tumor group was significantly shorter than that in the tumor-free group (37 *vs.* 396 days, $P = 0.000$) (Fig. 2). The median OS was significantly shorter in the BSC group than in the chemotherapy group (28 *vs.* 106 days, $P = 0.003$) (Fig. 3). Univariate analysis showed no significant correlations between OS and sex, age, histological category, bleeding symptoms, comorbidities or metastatic patterns.

**Table 2 Comparisons of clinicopathological characteristics between chemotherapy group and BSC group.**

| Clinicopathological factors | Chemotherapy (n=) (%) | BSC (n=) (%) | P value |
|---|---|---|---|
| Sex | | | 0.796 |
| Male | 12 (57.1) | 26 (52.0) | |
| Female | 9 (42.9) | 24 (48.0) | |
| Age (years) | | | 0.395 |
| <65 | 17 (81.0) | 35 (70.0) | |
| ≥65 | 4 (19.0) | 15 (30.0) | |
| Histological category | | | 0.053 |
| Highly differentiated | 3 (14.3) | 4 (8.0) | |
| Poorly differentiated | 14 (66.7) | 44 (88.0) | |
| Uncertain | 4 (19.0) | 2 (4.0) | |
| Tumor status | | | 0.156 |
| With tumor | 20 (95.2) | 40 (80.0) | |
| Tumor free | 1 (4.8) | 10 (20.0) | |
| Bleeding symptoms | | | 0.571 |
| Positive | 14 (66.7) | 37 (74.0) | |
| Negative | 7 (33.3) | 13 (26.0) | |
| Comorbidities | | | 0.792 |
| Negative | 9 (42.9) | 19 (38.0) | |
| Positive | 12 (57.1) | 31 (62.0) | |
| Metastatic patterns | | | 0.728 |
| Uncertain | 0 | 1 (2.0) | |
| Locoregional metastasis | 3 (14.3) | 9 (18.0) | |
| Hematogenous metastasis | 3 (14.3) | 4 (8.0) | |
| Peritoneal implantation | 1 (4.7) | 6 (12.0) | |
| Multiple metastatic patterns | 14 (66.7) | 28 (56.0) | |
| Negative | 0 | 2 (4.0) | |

Multivariate analysis showed that tumor status ($P = 0.000$) and treatment ($P = 0.000$) had independent effects on OS (Table 4).

## DISCUSSION

The incidence of DIC has been reported to be 1.6–1.7% in AGC patients (*Rhee et al., 2010*; *Takashima et al., 2010*). In this study, between June 2006 and March 2020, 14016 patients at Fujian Cancer Hospital were diagnosed with stomach cancer; DIC was detected in 105 of these individuals, for an incidence of 0.75%. This incidence rate is lower than that in the above studies. The reason for this difference is that our study calculated the incidence of DIC in AGC and EGC patients who could be treated surgically, while previous studies reported the incidence of DIC in AGC patients only. After excluding patients who were lost to follow-up, 71 patients were enrolled in this study. The histological category was

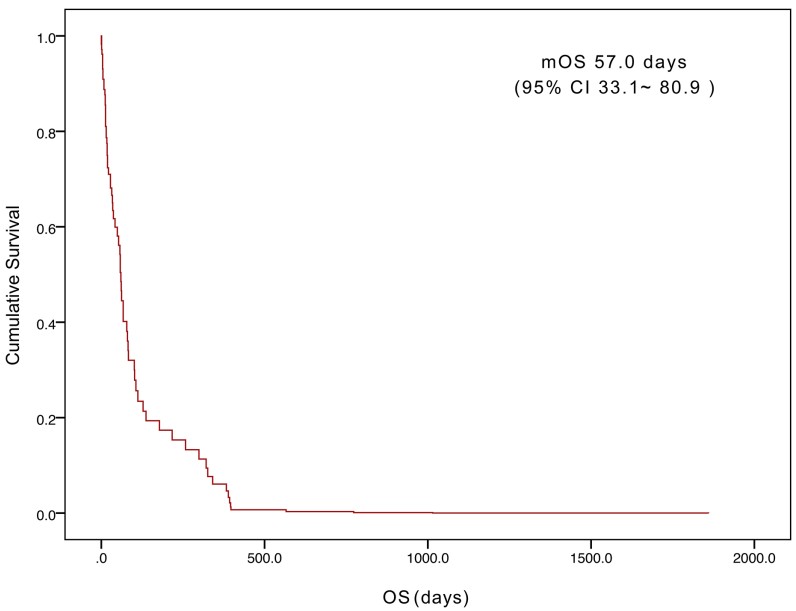

**Figure 1 Kaplan–Meier curves of overall survival for the 71 patients.**

poorly differentiated gastric cancer in 58 patients (81.7%), highly differentiated in seven patients (9.9%), and uncertain in six patients (8.4%). These data suggest that patients with poorly differentiated gastric cancer are more likely to develop DIC, and this result is similar to the findings of other studies (*Yeh & Cheng, 1998*; *Huang et al., 2008*). This suggests that more attention should be given to monitoring coagulation function in patients with poorly differentiated gastric cancer. Fifty-one patients (71.8%) had bleeding symptoms, with gastrointestinal bleeding as the most common. Forty-three (60.6%) patients had comorbidities at the time of DIC diagnosis, with simultaneous infection as the most common. Among the 71 patients, 42 (59.2%) had multiple metastatic patterns, which was the most frequently observed pattern. The above three findings suggest that gastrointestinal bleeding, multiple metastatic patterns and comorbidities at diagnosis with DIC are common in patients with gastric cancer complicated with DIC.

In the MAGIC study, individuals with operable illness who were given perioperative chemotherapy had a 36% 5-year survival rate (*Cunningham et al., 2006*). However, the median OS for AGC patients is less than one year, with a 5-year survival rate of approximately 5–20% (*Kamangar, Dores & Anderson, 2006*; *Cunningham et al., 2005*). In our study, the 5-year survival rate of 71 patients was 2.8%. The median OS was 57.0 days (95% CI: [33.1–80.9] days) in this study, which is consistent with previous studies (*Rhee et al., 2010*; *Hwang et al., 2014*). The results of multivariate analysis showed that tumor status ($P = 0.000$) and treatment ($P = 0.000$) had independent effects on OS. In this study, 11 patients (15.5%) developed DIC after curative gastric resection, at which time the patients had a tumor-free status. DIC was caused by severe infection and anastomotic leakage. In sepsis, DIC is associated with increased plasma D-dimer and decreased plasma fibrinogen, thrombin-antithrombin complex (TAT), and plasminogen activator inhibitor

**Table 3 Results of the univariate analysis to identify factors predicting the survival time of gastric cancer patients with DIC.**

| Clinicopathological factors | Median OS days [95% CI] | P value |
|---|---|---|
| Sex | | 0.152 |
| Male | 67.0 [33.8–100.2] | |
| Female | 37.0 [8.9–65.1] | |
| Age (years) | | 0.648 |
| <65 | 42.0 [14.9–69.1] | |
| ≥65 | 62.0 [49.2–74.8] | |
| Histological category | | 0.157 |
| Highly differentiated | 112.0 [96.6–127.4] | |
| Poorly differentiated | 37.0 [10.9–63.1] | |
| Uncertain | 67.0 [15.4–118.6] | |
| Tumor status | | 0.000 |
| With tumor | 37.0 [5.4–68.6] | |
| Tumor free | 396.0 [0.0–922.5] | |
| Bleeding symptoms | | 0.881 |
| Positive | 53.0 [27.0–79.0] | |
| Negative | 57.0 [17.6–96.4] | |
| Comorbidities | | 0.352 |
| Negative | 82.0 [36.6–127.4] | |
| Positive | 37.0 [10.0–64.0] | |
| Metastatic patterns | | 0.089 |
| Uncertain | 13.0* | |
| Locoregional metastasis | 83.0 [42.3–123.7] | |
| Hematogenous metastasis | 58.0 [55.4–60.6] | |
| Peritoneal implantation | 28.0 [0.0–61.3] | |
| Multiple metastatic patterns | 37.0 [10.0–64.0] | |
| Negative | 15.0Δ | |
| Treatment | | 0.003 |
| BSC | 28.0 [11.2–44.8] | |
| Chemotherapy | 106.0 [61.1–150.9] | |

Notes:
*Only one patient's metastatic pattern was uncertain.
Δ, Only two patients' metastatic pattern were negative.
CI, confidence interval.

type 1 (PAI-1) levels (*Koyama et al., 2014*). Although bacterial infections are the most prevalent cause of DIC, the underlying mechanisms are not completely understood. *Wang et al. (2019)* reported that outer membrane vesicles (OMVs), which are membrane-enclosed microvesicles released by a variety of bacteria, contribute to the pathogenesis of DIC during gram-negative bacterial infection. The possibility of postoperative DIC after tumor removal should be considered. Early monitoring and early detection and treatment of infection can reduce the occurrence of this type of DIC.

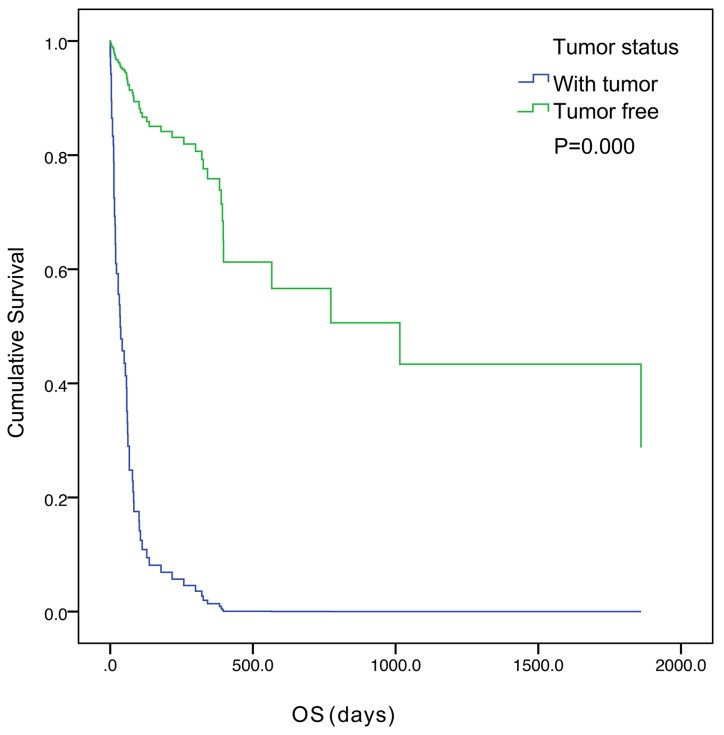

**Figure 2 Kaplan–Meier curves of overall survival were well separated according to tumor status.**

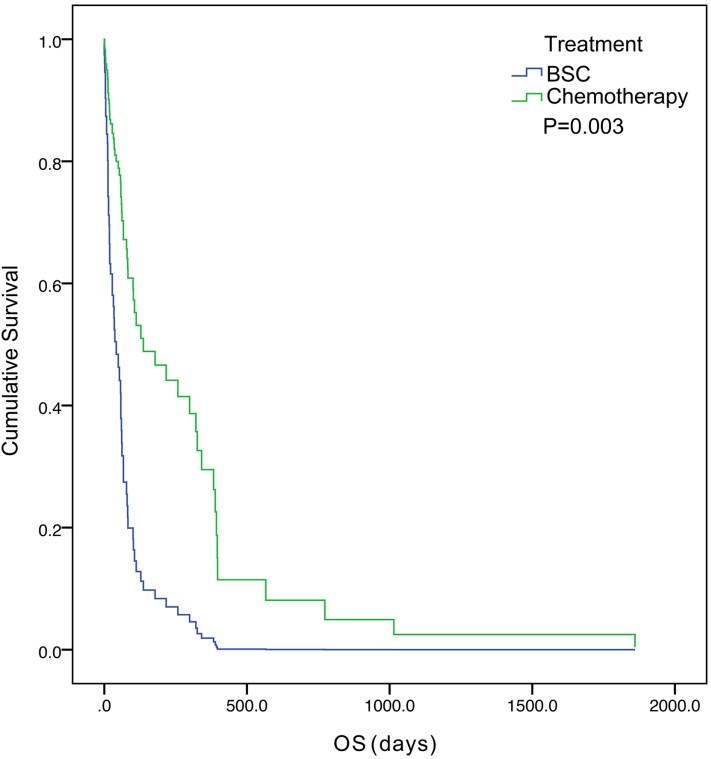

**Figure 3 Kaplan–Meier curves of overall survival were well separated according to treatment.**

**Table 4  Results of the multivariate analysis to identify factors predicting the survival time of gastric cancer patients with DIC.**

| Fators | B | SE | Wald | df | P | Exp(B) | 95.0% CI |
|---|---|---|---|---|---|---|---|
| Tumor status | −1.964 | 0.464 | 17.893 | 1 | 0.000 | 0.140 | [0.056–0.349] |
| Treatment | −1.020 | 0.288 | 12.529 | 1 | 0.000 | 0.360 | [0.205–0.634] |

Note:
B, regression coefficient; SE, standard error; df, degree of freedom; Exp(B), odds ratio; CI, confidence interval.

Through etiological treatment, such as the use of antibiotics, DIC can be corrected after achieving infection control, and patients can survive for a long time.

In this study, 60 patients (84.5%) had DIC at the initial presentation of gastric cancer or developed DIC as the tumor progressed during treatment. Patients with this type of DIC have a poor prognosis, and the majority of these patients die within 16–58 days (*Rhee et al., 2010*; *Hwang et al., 2014*). Among the 60 patients, only 20 (33.3%) patients received chemotherapy. This study showed that chemotherapy prolongs OS more than BSC. Our finding is consistent with the results of previous studies (*Rhee et al., 2010*; *Hwang et al., 2014*). Therefore, early chemotherapy should be considered in this population to improve their prognosis. In clinical practice, most of these patients have a poor performance status, severe thrombocytopenia and bleeding tendencies, meaning that chemotherapy would have high risks. Therefore, it is necessary to fully communicate the risks and possible benefits of chemotherapy with the patients and their families before the start of treatment. Paclitaxel every week was the most common chemotherapy regimen, with 10 of 21 (47.6%) patients receiving this regimen. For patients with a poor performance status (PS) and high risk for chemotherapy, we consider single-agent chemotherapy to be a safe and efficient treatment option. After the patient's physical condition improves, combination with other chemotherapy drugs can be considered to improve the curative effect. This opinion has also been confirmed in some studies. *Yeh & Cheng, 1998* have suggested that 5-FU and leucovorin (HDFL) is an ideal initial chemotherapy regimen for gastric cancer patients with DIC. According to *Hwang et al. (2012)*, first-line single-agent palliative chemotherapy showed relatively good clinical efficacy in AGC patients with a poor PS. *Wu et al. (2019)* reported that liposome-paclitaxel was given to AGC patients with a poor PS and that S-1 was also given after improvement of the patient's physical condition. These outcomes demonstrate that this treatment can extend patients' OS and progression-free survival (PFS), with manageable toxicity.

Data from one center were retrospectively analyzed in this research, and data collected from medical records has limitations. We enrolled 71 gastric cancer patients with DIC. This sample is still considered small although it is larger than those in earlier studies that only assessed DIC in AGC patients. Our patients included 60 AGC patients and 11 EGC patients. To obtain more reliable results in the future, prospective studies using larger patient cohorts that collect more detail about patients' clinicopathologic characteristics and treatment are needed.

## CONCLUSIONS

Gastrointestinal bleeding, multiple metastatic patterns and comorbidities at diagnosis with DIC are common in patients with gastric cancer complicated with DIC. Patients with poorly differentiated gastric cancer are more likely to develop DIC. This suggests that more attention should be given to monitoring coagulation function in patients with poorly differentiated gastric cancer. Treatment and tumor status are separate risk variables for survival in gastric cancer patients with DIC. DIC patients without tumors have a good prognosis and can be cured by appropriate etiological correction and symptomatic treatment. Chemotherapy can improve the prognosis of DIC patients with tumors.

## ACKNOWLEDGEMENTS

We would like to thank all our colleagues from the Department of Medical Oncology, Clinical Oncology School of Fujian Medical University, Fujian Cancer Hospital for their support.

### Funding

This work was supported by the Fujian Provincial Clinical Research Center for Cancer Radiotherapy and Immunotherapy (2020Y2012) and the Fujian Provincial Health Technology project (2020J011111). The funders had no role in study design, data collection and analysis, decision to publish, or preparation of the manuscript.

### Grant Disclosures

The following grant information was disclosed by the authors:
Fujian Provincial Clinical Research Center for Cancer Radiotherapy and Immunotherapy: 2020Y2012.
Fujian Provincial Health Technology: 2020J011111.

### Competing Interests

The authors declare that they have no competing interests.

### Author Contributions

- Ling Chen conceived and designed the experiments, performed the experiments, analyzed the data, prepared figures and/or tables, authored or reviewed drafts of the article, and approved the final draft.
- Jing Lin conceived and designed the experiments, performed the experiments, analyzed the data, prepared figures and/or tables, authored or reviewed drafts of the article, and approved the final draft.
- Yu Chen conceived and designed the experiments, performed the experiments, prepared figures and/or tables, and approved the final draft.
- Jiami Yu performed the experiments, prepared figures and/or tables, and approved the final draft.

- Xiaojie Wang performed the experiments, prepared figures and/or tables, and approved the final draft.

## Human Ethics

The following information was supplied relating to ethical approvals (*i.e.*, approving body and any reference numbers):

The study was approved by the Ethical Committee of Clinical Oncology School of Fujian Medical University, Fujian Cancer Hospital (permit number K2022-133-01).

## Data Availability

The raw data are available in the Supplemental File.

## Supplemental Information

Supplemental information for this article can be found online at http://dx.doi.org/10.7717/peerj.16527#supplemental-information.

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
