# Peer review of "Clinicopathologic features and prognosis of 71 patients with gastric cancer and disseminated intravascular coagulation"

_PeerJ, doi:10.7717/peerj.16527_

## Round 0.1 · original submission · Major Revisions

All three reviewers have given suggestions for revision, and the author is requested to revise them carefully.

·

Basic reporting

The authors analyze the clinicopathologic features and identify factors for the survival time of gastric cancer patients with DIC.

Experimental design

1. What are the details of treatments for gastric cancer patients with DIC receive?
2. What’s the difference of aggressive chemotherapy with other therapies?
3. What’s the significance to use univariate analysis to identify factors predicting the survival time of gastric cancer patients with DIC?

Validity of the findings

1. What’s the characteristics and five-year survival rate of gastric cancer?
2. What’s the molecular mechanism of diffused intravascular coagulation (DIC)?
3. Which genes are involved in DIC formation?

Reviewer 2 ·

Basic reporting

no comment

Experimental design

no comment

Validity of the findings

no comment

Additional comments

1, The prognosis is poor for those who had DIC at the time of the initial onset of gastric cancer or developed DIC as the tumor progressed during treatment. The patient's general condition is expected to be poor. Chemotherapy has risks. One way to think of this is that chemotherapy did not prolong survival, but rather that patients who had the choice to receive chemotherapy (because they had the stamina and support from their families) lived longer. Please provide more details about patients who did not receive chemotherapy.
2, Multiple chemotherapy methods are being used, and the effectiveness of chemotherapy seems ambiguous. Please further emphasize the evidence that single-agent chemotherapy is a safe and effective treatment option.
3, The conclusion is suitably cautious about the findings. It would be helpful to discuss how these results compare with those of other similar studies. Also, more specific suggestions for future research could strengthen the conclusion.

Reviewer 3 ·

Basic reporting

This manuscript focuses on the clinicopathological characteristics and prognosis analysis of gastric cancer patients with DIC, which is innovative, but some still need to be improved.
1. There is no clear expression of the significance of this study in the background part. I think the first paragraph of the discussion section (lines 180-185) can be placed in the background section, which can better reflect the innovation of the manuscript.
2. There is a self-contradictory phenomenon between the background and the abstract (e.g., line 37 “DIC is rare in patients with stomach cancer.” and line 73 “DIC has been frequently found in gastric cancer.”).
3. English writing should be improved.
4. The title of this manuscript is “Clinicopathologic features and prognosis of 71 patients with gastric cancer and disseminated intravascular coagulation”, but the conclusion section does not express and summarize clinicopathological features.
5. There are few references, especially the background part, and the data analysis part, as to why this method of data analysis was used and what advantages it has over other analytical methods of the same type.

Experimental design

no comment

Validity of the findings

It is necessary to supplement some data, such as the results of single factor analysis.

Additional comments

The clarity of the image needs to be improved. The contrast between the two groups of data is not obvious in color, and the resolution of the image should be improved.

---

## Round 0.2 · accepted · Accept

Although there are still two reviewers who did not review the revised manuscript of the author, the initial review comments of the above two reviewers are minor revisions. In the revised draft, the author also seriously answered their questions. After reviewing the manuscript, I have concluded that there is no significant risk of publication, and in order to save the author's time, I agree to publish it.

·

Basic reporting

The manuscript used clear and professional English to illustrated the scientific issues.

Experimental design

The research questions have been well defined according to the suggestions from the reviewers.

Validity of the findings

All the data are statistically sound and controlled. Conclusions are well stated.